# The Role of Bone Marrow Mesenchymal Stem Cell Derived Extracellular Vesicles (MSC-EVs) in Normal and Abnormal Hematopoiesis and Their Therapeutic Potential

**DOI:** 10.3390/jcm9030856

**Published:** 2020-03-20

**Authors:** Aristea K. Batsali, Anthie Georgopoulou, Irene Mavroudi, Angelos Matheakakis, Charalampos G. Pontikoglou, Helen A. Papadaki

**Affiliations:** 1Haemopoiesis Research Laboratory, School of Medicine, University of Crete, 71003 Heraklion, Greece; tea_ios@yahoo.gr (A.K.B.); anthieg87@yahoo.com (A.G.); eirimav@gmail.com (I.M.); a.matheakakis@gmail.com (A.M.); xpontik@uoc.gr (C.G.P.); 2Department of Hematology, School of Medicine, University of Crete, 71003 Heraklion, Greece

**Keywords:** mesenchymal stem cells (MSCs), extracellular vesicles (EVs), MSC-EVs, exosomes, micro-vesicles (MVs), hematological malignancies, hematopoietic stem cell transplantation (HSCT)

## Abstract

Mesenchymal stem cells (MSCs) represent a heterogeneous cellular population responsible for the support, maintenance, and regulation of normal hematopoietic stem cells (HSCs). In many hematological malignancies, however, MSCs are deregulated and may create an inhibitory microenvironment able to induce the disease initiation and/or progression. MSCs secrete soluble factors including extracellular vesicles (EVs), which may influence the bone marrow (BM) microenvironment via paracrine mechanisms. MSC-derived EVs (MSC-EVs) may even mimic the effects of MSCs from which they originate. Therefore, MSC-EVs contribute to the BM homeostasis but may also display multiple roles in the induction and maintenance of abnormal hematopoiesis. Compared to MSCs, MSC-EVs have been considered a more promising tool for therapeutic purposes including the prevention and treatment of Graft Versus Host Disease (GVHD) following allogenic HSC transplantation (HSCT). There are, however, still unanswered questions such as the molecular and cellular mechanisms associated with the supportive effect of MSC-EVs, the impact of the isolation, purification, large-scale production, storage conditions, MSC source, and donor characteristics on MSC-EV biological effects as well as the optimal dose and safety for clinical usage. This review summarizes the role of MSC-EVs in normal and malignant hematopoiesis and their potential contribution in treating GVHD.

## 1. Introduction

The bone marrow (BM) is a dynamic organ composed of hematopoietic stem cells (HSCs), their progeny, endothelial cells, and cells originating from mesenchymal stem cells (MSCs) such as osteoblasts and adipocytes. These cells, in association with the extracellular matrix, organize a specialized microenvironment regulating the formation of mature hematopoietic cells and their proper function.

MSCs reside throughout the body and are characterized by the capacity to self-renew and differentiate into several mesoderm lineages such as chondrocytes, osteocytes, and adipocytes in vivo [1,2]. Many reports have demonstrated that MSCs can differentiate into non-mesodermal and endodermal lineages in vitro [2,3]. They can be isolated from the BM, adipose tissue, dental pulp, Wharton’s jelly, cervical tissue, placentae, muscle tissue, lung, synovial membranes, and umbilical cord (UC) blood [4,5]. MSCs are Cluster of Differentiation-73 (CD73), CD90, and CD105 positive and CD45, CD34, CD14, CD19, and human leukocyte antigen-DR isotype (HLA-DR) negative [6].

The main role of MSCs is the support, maintenance, and regulation of HSCs’ properties [7]. The interaction between these two cell types results in the prevention of HSC differentiation and protection from apoptosis, which promotes self-renewal and maintenance of stemness [2,8]. Moreover, in cases where immune responses are excessive, MSCs can suppress T cells, B cells, macrophages, dendritic cells, and natural killer (NK) cells [9,10]. This immunomodulatory effect of MSCs is mainly mediated by producing different bioactive molecules such as adhesion molecules (intercellular adhesion molecule-1, ICAM-1, activated leukocyte cell adhesion molecule, ALCAM), growth factors (epidermal growth factor, EGF, transforming growth factor beta, TGF-β, granulocyte-macrophage colony-stimulating factor, GM-CSF), cytokines (inteleukin-IL-1α, IL1β, IL6, and IL8) angiogenic factors (Vascular Endothelial Growth Factor, VEGF; Platelet-Derived Growth Factor, PDGF) and immunomodulatory molecules (prostaglandin E2 (PGE2); human leukocyte antigen G (HLA-G) [11,12,13]. All these molecules are responsible for the paracrine effects of MSCs on neighboring cells [14,15]. The ability of MSCs to inhibit immune cell proliferation, to induce regulatory T/B cells (T/Bregs) lymphocyte proliferation, to mediate dendritic maturation, and to migrate to injured tissues for regenerative purposes are the main reasons that MSCs have been widely used in many clinical trials for treating immune-mediated disorders such as Graft-Versus-Host Disease (GVHD), multiple sclerosis, arthritis, sepsis, asthma, and dermatitis [16,17,18,19]. On the other hand, MSCs may be involved in the tumorigenesis process of hematological malignancies similar to their previously described effects in other cancers i.e., breast cancer, neuroblastoma, osteosarcoma, and adenocarcinomas [7].

The study of MSCs in normal and disease states frequently displays low reproducibility that might be due to differences in the isolation methods and culture conditions [20,21]. Therefore, there is currently an increasing interest in the investigation of the isolation procedures, the biophysical characteristics, and biological and clinical effects of MSC-derived extracellular vesicles (MSC-EVs) that mimic MSC properties [22,23]. Similar to MSCs, MSC-EVs are involved in cell proliferation and differentiation, antigen presentation, angiogenesis, and demonstrate anti-inflammatory and regenerative properties [24,25].

## 2. Extracellular Vesicles

EVs are a heterogeneous group of bilayer membrane structures that, according to their size, shape, biogenesis, and composition, are classified into two major categories known as exosomes and micro-vesicles (MVs) [26]. Exosomes are particles of endosomal origin with sizes ranging from 40 to 100 nm in diameter. They are generated from the internal budding of multivesicular bodies and released via exocytosis [27]. MVs are exosome-like vesicles with a size ranging from 50 to 1000 nm in diameter that are released by the budding of the cell membrane [28].

EVs are secreted by a variety of cell types such as MSCs, T cells, B cells, and dendritic cells, and can be isolated from all biological body fluids including serum, blood, breast milk, urine, and semen [29]. They are positive for CD29, CD44, CD73, and CD105 similar to MSCs, but they are also positive for CD81, CD82, CD63, CD53, CD9, and CD37 due to their endosomal origin [30]. EVs can be transported through blood and biological body fluids due to their size and interact with many target cells via surface receptors. Based on their cellular origin and biogenesis, EVs contain different proteins, soluble factors, and microRNAs. The biological effect of EVs is exerted in target cells via both an endocrine effect on distant cells and a paracrine effect on adjacent cells [31]. They are internalized via endocytosis, phagocytosis, pinocytosis, and membrane fusion [32]. Besides their content, EVs’ biological effect depends on the functional and metabolic condition of the recipient cells [33].

The secretion of EVs from MSCs is influenced by inflammatory stimuli, stress, high levels of intracellular calcium, and acidic pH both in physiological and pathological conditions [34,35]. In this aspect, several studies have shown that MSC-EVs might be used as a cell-free therapy for cardiovascular, liver and kidney diseases, immunological disorders, wound healing, and tumor inhibition [27,36]. Moreover, EVs are safer as compared to their parental cells because they display low immunogenicity and no tumorigenicity. Additionally, they lack the risk for aneuploidy due to non-self-renewal ability [30]. The isolation, storage, and administration of EVs is much more cost-effective than MSCs.

EVs’ ability to transport molecules and to target specific cell populations raises possibilities for their development as pharmaceutical vehicles and sources of diagnostic and prognostic markers. As mentioned above, it is well recognized that cells release two EV-subtypes with different mechanisms of biogenesis and organelle origin [26,27]. If functional differences between EV-subtypes do exist, highly purified vesicle populations are an absolute necessity for the development of EV-based therapies. Sample collection, isolation, and purification of EVs are linked concepts. Therefore, position papers and statements from the International Society for Extracellular Vesicles (ISEV) should be consulted for the most updated guidance on isolation and characterization [37,38]. Similar to the case for cellular therapeutics, EV production on an industrial scale must eventually occur in serum-free conditions to exclude xenogeneic components, and the cellular source must be considered carefully. Since removing serum may change the phenotypic and functional characteristics of cells and the EVs they produce, a culture change would necessitate confirmation that EVs’ properties remain the same across media [39,40].

Current practices are to use either chemically-defined media or human platelet lysate (HPL) as a serum replacement. Chemically-defined media may allow a better control of production conditions, which are crucial for industrial-scale manufacturin. Although HPL is already used for the production of functionally efficient cellular therapeutics [41,42], it may contain unidentified pathogenic components that, theoretically, might be spread and hamper global up-scaling strategies [43].

Since EVs may reflect molecular expression patterns and functions of their parental cells, the optimal EV isolation method depends on the intended therapeutic use, route of administration, starting material (milk, plasma, urine, or cell culture), and desired end product (MVs, exosomes or total EVs) [44,45]. In view of the use of EVs as therapeutics agents, it is necessary to develop scalable, reproducible, and good manufacturing practice (GMP)-compliant manufacturing protocols in the context of appropriate regulatory frameworks. For example, the US Food and Drug Administration (FDA) (https://www.fda.gov/drugs) recommends that the route of administration, the acute and repeated dose, the local toxicity studies with histological evaluation, and the route-specific considerations should be taken into consideration when designing EV-based clinical trials [46].

## 3. MSC-EVs in Normal and Abnormal Hematopoiesis

The hematopoietic niche and its components construct a biological environment that maintains the homeostasis and responds to stress, damage, or disease conditions. These processes are mediated by chemokines, cytokines, growth factors, metabolites, multiple signaling pathways, and MSC-EVs [47]. EVs that target different cell types and regulate their fate are involved both in normal hematopoiesis and in hematological malignancies (Figure 1).

### 3.1. Roles of MSC-EVs in Normal Hematopoiesis

MSC-EVs may contribute to the activation of quiescent HSCs following different stimuli such as hemorrhage, changes in oxygen concentration, chemotherapy, and irradiation [48]. In vitro studies have demonstrated that MSC-MVs may enhance the proliferation of UC blood CD34^+^ HSCs to a lesser degree compared to their cellular counterparts by inducing the Wnt/β-catenin signaling pathway that increases the proliferation and inhibits HSC differentiation [7]. Moreover, miRNAs of MSC-EVs induce cell survival and proliferation and inhibit apoptosis or differentiation of all hematopoietic lineages [30]. In a co-culture system, the MSC-EVs miRNAs have been reported to increase the migration of CD34^+^ HSCs from peripheral blood (PB) to BM niche, via an increase in Cysteine-X-cysteine (CXC) motif chemokine receptor type 4 (CXCR4) expression. However, another study showed that MSC-EVs increase the differentiation of HSCs to myeloid progenitors by interacting with Toll-like receptor 4 (TLR4) [49]. MSC-EVs were able to decrease radiation injury of murine HSCs by stimulating their proliferation in vitro [50].

MSC-EVs have both pro-angiogenic and anti-angiogenic properties. UC-derived MSCs secrete EVs that promote angiogenesis and MSC proliferation and migration by activating the Wnt/β-catenin signaling pathway [51]. Under hypoxic conditions, EVs highly express VEGF, VEGF Receptor 2 (VEGFR2), angiogenin, and IL6. However, MSC-EVs act on endothelial cells by transfering proangiogenic miR424, miR30c, and miR30b molecules [52].

MSC-EVs also have immunomodulatory effects akin to their parental cells. They affect the proliferation, polarization, maturation, and migration of macrophages via the production of chemokines, growth factors, and other signaling molecules [53]. MSC-EVs are also involved in the disturbance of macrophages 1/macrophages 2 (M1/M2) balance observed in macrophage populations in many pathological conditions by decreasing the expression of proinflammatory signals such as tumor necrosis factor (TNF)-α and IL6 from M1 macrophages and enhancing the expression of Arginase (Arg1) from M2 macrophages [54]. Moreover, the immunoregulatory activity of MSC-EVs is also exerted by impairing dendritic cell maturation, activating neutrophils, inhibiting NK cell proliferation, suppressing B and T cell proliferation, and elevating the number of T regulatory cells [55].

Specific effects of BM-derived MSC-EVs (BM-MSC-EVs) in the homeostasis and maintenance of the BM microenvironment are mediated through abundant miRNA secretion including miR143, miR10b, miR22, miR486, and miR21 [56,57]. More specifically, miR143 has immunomodulatory functions, mir10b and miR22 regulate the differentiation of MSCs, and miR486 and miR21 are involved in MSC proliferation and angiogenic activity [56] (Table 1).

### 3.2. Roles of MSC-EVs in Hematological Malignancies

The BM microenvironment has been reported to affect the proliferative, self-renewal, and migratory properties of HSCs in a variety of hematological malignancies including myelodysplastic syndromes (MDS), multiple myeloma (MM), and acute myeloid leukemia (AML), among others, which leads to disease onset and/or progression [68]. MSCs in these diseases display altered properties such as growth defects, accelerated senescence, dysregulated osteogenic differentiation, genomic instability, and compromised capacity to support normal hematopoiesis [69,70,71,72]. Apart from the abnormal intrinsic properties, MSCs may influence the BM microenvironment via paracrine mechanisms by secreting soluble factors including EVs [73].

The role of MSC-EVs has been extensively studied in MM. It has been shown that MSC-EVs from healthy donors inhibit the growth of MM cells while their counterparts from MM patients promote the tumor growth [56]. MSC-EVs from MM patients express high levels of IL6, CCL2, and fibronectin and low levels of the tumor suppressor mir15a, which is capable of inhibiting MM cell growth but also to induce apoptosis, which maintains the disease in a stable state [59]. Another study has shown that MSC-derived exosomes activate the AKT pathway and inhibit the p38, p53, and c-Jun N-terminal kinase (JNK) pathway. Therefore, this promotes the survival and proliferation of MM cells [60]. Furthermore, the viability, proliferation, and migration of MM cells has been reported to decrease following treatment with BM-MSC-MVs from healthy donors via the activation of a mitogen-activated protein kinases (MAPK) pathway in contrast to MVs from MM patients [61]. It has also been reported that MSC-derived exosomes from old healthy donors display weaker immunomodulatory effects on MM cell lines in contrast to exosomes from younger donors due to the loss of young BM-MSC exosome-specific miRNAs. It is known that these miRNAs are related to cancer as the development of the majority of cancers is considered to be related to age [62]. Other studies suggest that MM EVs block the differentiation and mineralization of BM-MSCs [64,65].

With regard to AML, it has been reported that exosomes from AML cells contain miR-155, miR-375, and miR-150. These miRNAs regulate the secretion of cytokines and growth factors by BM micro-environment cells as well as the proliferation and migration of HSCs by decreasing CXCR4 expression and affecting the CXCR4/SDF-1 axis, which is fundamental for the retention and differentiation of HSCs in the BM [74]. Additionally, MSC exosomes from AML patients have been reported to protect leukemic cells carrying the fms like tyrosine kinase 3 (FLT3) internal tandem duplication from treatment with the AC220 specific FLT3 inhibitor [65]. Another study demonstrated that AML exosomes promote the leukemic cell survival and proliferation and suppress normal hematopoiesis [66].

Many studies have shown that BM-MSCs are involved in the generation of dysplastic hematopoietic cells, which contribute to disease initiation and evolution [67,75]. Pavlaki et al. have shown a decreased gene expression of the cyclin-dependent kinase inhibitors CDKN1A, CDKN2A, and CDKN2B in BM-MSCs in patients with MDS and an impaired capacity to promote the differentiation of CD34^+^ cells to myeloid and erythroid lineage. Moreover, BM-MSCs from MDS patients appeared to have a defective osteogenic and adipogenic capacity due to a downregulated canonical WNT (Wnt/β-catenin) expression and upregulated canonical WNT inhibitors [76]. BM-MSCs from MDS-patients are genetically unstable, but this did not affect their proliferative or survival capacity [77]. Another study has shown that there are no differences in the number, the differentiation potential, and the gene expression of proinflammatory or growth-promoting cytokines between the BM-MSCs from MDS patients and healthy donors [78]. However, little is known about the role of MSC-EVs in this disease. Τo the best of our knowledge, there is one study showing that MSC-MVs from MDS patients modify CD34^+^ cell properties, promote cell viability and clonogenic capacity, and alter their miRNA and gene expression [67].

## 4. MSC-EVs and Graft Versus Host Disease

Although targeted therapies have improved the prognosis of patients with hematological diseases, allogeneic hematopoietic stem cell transplantation (HSCT) remains the only curative therapy for patients with negative prognostic factors and/or refractory disease. GVHD occurs when immune cells transplanted from a genetically non-identical donor recognize recipient allo-antigens, which leads to their activation and, subsequently, organ damage [79].

Various immunosuppressants have been clinically applied for the prevention and treatment of GVHD [80]. However, a substantial proportion of HSCT recipients will likely develop this potentially life-threatening complication. The efficacy of standard primary therapy with corticosteroids is about 50%, and the complete response rate to secondary therapy with a variety of immunosuppressants is about 30% with a median overall survival of less than one year in steroid-refractory patients [81].

A series of clinical studies have examined the efficacy of systemic infusion of culture-expanded BM-MSCs for acute GVHD in allogeneic HSCT-treated patients and the results have shown overall responses ranging from 30% to 80% [82]. Attempts to improve the outcome of BM-MSC therapy have been based on the concept that this form of therapy is dependent on the number of infused cells that can successfully traffic to sites of damaged or diseased tissues. However, systemic administration of an increased number of cells has not augmented the therapeutic effects of BM-MSCs in GVHD [83,84]. In addition, in a study in which BM-MSCs were directly delivered into the gut via the mesenteric artery, the outcome was not more effective than a systemic injection [85]. These clinical results underscore the current hypothesis that the therapeutic effects of BM-MSCs, at least in acute GVHD, are attributed mainly to secreted immune-modulatory factors.

Kordelas, L. et al. showed, for the first time, that infusion of MSC-derived exosomes may significantly improve the symptoms of the steroid-resistant acute GVHD shortly after administration and without significant side effects [86]. Although an in vivo and ex vivo decrease of pro-inflammatory cytokines such as TNF-α, IL1β, and interferon (IFN)-γ was observed following exosomes’ infusion, the biologic mechanisms by which BM-MSC-EVs exert their functions and effects remain unknown. Moreover, it has been reported that the therapeutic effect of BM-MSC-derived EVs to GVHD is associated with the inhibition of T cell induction and preservation of circulating naïve T cells [87].

It has also been shown that systemic infusion of human BM-MSC-EVs prolonged the survival of mice with acute GVHD and reduced the pathologic damage in multiple GVHD-targeted organs [88]. In EV-treated GVHD mice, CD4^+^ and CD8^+^ T cells were suppressed. Moreover, BM-MSC-EVs seemed to preserve CD4^+^CD25^+^Foxp3^+^ regulatory T cell populations. On the contrary, normal human dermal fibroblasts-derived EVs did not ameliorate the clinical or pathological characteristics of acute GVHD in mice, which suggests an immunoregulatory function unique to BM-MSC-EVs.

According to an array analysis, multiple soluble factors associated with the amelioration of GVHD are highly expressed in BM-MSC-EVs, such as CXCL12 [89]. A recent study showed that BM derived MSC-exosomes can effectively ameliorate chronic GVHD in mice by inhibiting the activation and infiltration of CD4^+^ T cells. Furthermore, MSC-exosomes exhibit immunomodulatory potential by inducing regulatory T cells and inhibiting Th17 cells. These immunosuppressive effects of MSCs-exosomes are mediated, at least in part, through IL-17α and IL-21 [90].

Until now, only one study has indicated that human UC-derived MSC-EVs (UC-MSC-EVs) represent an ideal alternative in the prophylaxis of acute GVHD in a mouse model of allogeneic HSCT by modulating immune responses. Recipients treated with UC-MSC-EVs had a significantly lower number of CD3^+^CD8^+^ T cells, reduced serum levels of IL2, TNF-α, and IFN-γ, a higher ratio of CD3^+^CD4^+^/CD3^+^CD8^+^ T cells, and higher serum levels of IL10. An in vitro experiment demonstrated that UC-MSC-EVs inhibited the mitogen-induced proliferation of splenocytes in a dose-dependent manner and the cytokine changes were similar to those observed in vivo [91].

## 5. Conclusions

A number of biologic functions and effects of MSCs has been shown to be mediated by their EV derivatives. MSC-EVs may influence the HSC and their micro-environment in normal and disease states by transferring their content and mediating anti-malignant or pro-malignant effects through largely unknown and still controversial mechanisms. Beyond the potential roles in the physiology and pathophysiology of hematopoiesis, several studies have also investigated the use of MSC-EVs as potential alternatives to MSCs for improving HSCs expansion and engraftment and for preventing GVHD following HSCT. It is anticipated that the MSC-EV usage may have potential advantages over their cellular counterparts in some aspects. For example, in contrast to MSCs, MSC-EV infusion will not be complicated by the possibility of being trapped in lung capillaries, which might reduce their approach in the damaged tissues/organs and the effectiveness of the treatment nor by the danger of malignant transformation associated with MSC infusion. However, there are still unanswered questions in the pre-clinical level such as the molecular and cellular mechanisms associated with the supportive action of MSC-EVs, the impact of the isolation protocols, the effect of MSCs’ source and donor characteristics (i.e., gender and age) on EV biological characteristics and bioactive cargoes, and more [92,93]. Furthermore, the field is open for clinical studies investigating the potential changes of HSCs following MSC-EV infusion, the optimal dosage and safety of MSC-EVs, and the effect of the collection, purification, large-scale production, and storage on patients’ outcomes.

## Figures and Tables

**Figure 1 jcm-09-00856-f001:**
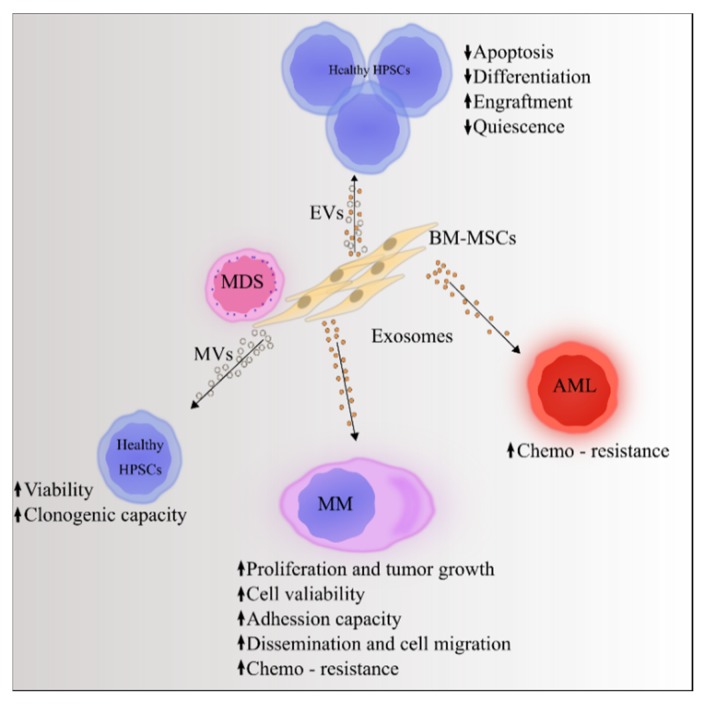
Role of extracellular vesicles in normal and abnormal hematopoiesis. Extracellular vesicles from BM-MSCs inhibit apoptosis and differentiation of HSPCs and induce engraftment of precursor cells in the BM niche. BM-MSC-derived exosomes from patients with multiple myeloma increase cell proliferation, viability, and chemo-resistance and facilitate tumor growth and cell migration. BM-MSC-derived exosomes from patients with Acute Myeloid Leukemia modulate chemo-resistance in cell lines. BM-MSC-derived micro-vesicles from patients with low-risk myelodysplastic syndromes increase cell viability and clonogenic capacity of CD34^+^ cells from healthy donors. Abbreviations: AML: acute myeloid leukemia, BM-MSCs: bone marrow mesenchymal stromal cells, EVs: extracellular vesicles, HSPC: hematopoietic stem and progenitor cells, MDS: myelodysplastic syndromes, MM: multiple myeloma, MVs: microvesicles, HPSCs: human hematopoietic stem and progenitor cells.

**Table 1 jcm-09-00856-t001:** hBM-MSCs derived extracellular vesicles’ (EVs) effects on healthy hematopoiesis and hematological malignancies.

Normal or Disease State	Source of MSC-EVs	EV Type	Biological Effect	Mechanism	Reference
Normal Hematopoiesis	BM-MSCs from healthy donors	all EVs	BM-MSCs-EVs reduced apoptosis, inhibited differentiation of target cells in vitro and increase engraftment of CD34^+^ umbilical cord blood cells in in vivo murine BM	Differential gene regulation comprising small RNA-target genes in CD34^+^ cells exposed to BM-MSC-EVs compared with naive CD34^+^ cells. Reduced caspase 3/7 activity, down-regulation of MPL and ZFP36 and up-regulation of chemotactic factors (IL1b, CSF2, CCL3, GATA2, and CXCR4) were the main molecular mechanisms	[49]
Normal Hematopoiesis	BM-MSCs and adipose-derived MSCs from C57BL/6 mice	all EVs	MSC-EVs prompt a loss of HSPC quiescence and expansion of myeloid biased lineage	Exosomes engaged TLR-4 followed by NF-κB upregulation that led to downstream activation of Hif-1 and CCL2 target genes and increased secretion of pro-inflammatory cytokines	[58]
MM	BM-MSC from patients with MM, Smoldering MM, MGUS, healthy donors, and a human stromal cell line HS-5	Exosomes	HD-BM-MSCs-exosomes reduced MM cell proliferation while MM-BM-MSCs’ exosomes increased MM cell proliferation in vitro and increased cell adhesion capacity MM-BM-MSCs-exosomes promoted tumor growth and dissemination while HD-MSCs-exosomes inhibited tumor growth in an in vivo setting	Differential miRNA and protein transfer	[59]
MM	BM-MSC from MM patients and healthy donors	Exosomes	Both MM-BM-MSCs- and normal-MSCs-exosomes induced drug resistance to Bortezumib in MM cells	Both MM and normal-BM-MSCs-exosomes activated chemotaxis (CXCR4, SDF-1-mediated, and MCP-1-mediated pathways), increased anti-apoptotic proteins (Bcl-2), and inhibited the activation of caspase-9 and caspase-3. Additionally, exosomes of both sources altered phosphorylation of p38, p53, and JNK as well as prevented the inhibition of AKT pathway. Exosomes managed to inhibit reduction of Bcl-2 caused by Bortezomib	[60]
MM	BM-MSC from MM patients and healthy donors	MVs	MM-MSCs-MV increased viability, proliferation, migration capacity, and translational activity of MM cells compared with HD-MSCs-MVs treated MM cells	MM-MSCs-MVs increased phosphorylation of MAPKs (pERK1/2 and pJNK) and activation of TI factors (peIF4E and peIF4GI) in MM cells compared with HD-MSCs- MVs	[61]
MM	BM-MSCs from healthy donors	Naïve exosomes Exosomes transfected with miR340 and miR365 mimics	BM-MSCs’-exosomes from younger donors inhibited angiogenic response of MM-HR cells compared with BM-MSCs’-exosomes from older donors and control miR340-enriched exosomes inhibited angiogenesis and proliferation of MM-HR cells	miR340 enriched exosomes suppressed cMET translation	[62]
MM	5TGM1 cells and C57BL6/KalwRij mouse model	Small EVs	EVs enhance the osteoclast activity and block the osteoblast differentiation in vitro and in vivo	The blockage of secreted exosomes with sphingomyelinase inhibitor GW4869 increase the cortical bone volume and sensitize the myeloma cells to bortezomid	[63]
MM	BM-MSCs from MM patients and healthy donors	Exosomes	MM-BM-MSCs-exosomes promote the secretion of IL-6 and suppress the osteoblastic differentiation and mineralization of BM-MCs	MM-BM-MSCs-exosomes increase the expression of APE1 and NF-kB and decrease the expression of Runx2, Osterix, and OCN	[64]
AML	BM-MSCs from AML patients and healthy donors	Exosomes	BM-MSCs-exosomes increased chemo-resistance to Cytarabine (for both exosome sources) and Quizartinib (only for AML-BMSCs-exosomes) of AML cells		[65]
AML	AML cell lines and C57BL/Ka (B6), C57BL/Ka-Thy1.1-CD45.1, B6-Rag2^−/−^γc^−/−^, and NOD-SCID-γc^−^/^−^ mouse models	Exosomes	AML-exosomes promote the leukemic cell survival and proliferation and suppress normal hematopoiesis	AML-exosomes induce the expression of DKK1, a suppressor of normal hematopoiesis and osteogenesis. AML-exosomes reduce the ability of BM cells to support normal hematopoiesis by downregulating CXCL12, KITL, and IGF1 (hematopoietic stem cell supporting factor)	[66]
MDS	BM-MSC from MDS patients and healthy donors	MVs	MDS-BM-MSCs-MVs increased viability and clonogenic capacity of CD34^+^ compared with untreated cells	Downregulation of MDM2 protein expression in CD34^+^ cells after exposure to MDS-MVs	[67]

**Abbreviations:** AKT: Protein kinase B. AML: Acute Myeloid Leukemia. Bcl-2: B-cell lymphoma 2 gene. APE1: Apurinic/Apyrimidinic Endonuclease 1. hBM: human bone marrow. B6: C57BL/6 strain. B6-Rag2^−/−^: B6 recombination-activating gene–deficient. MSCs: mesenchymal stem cells. EVs: extracellular vesicles. BM-MSCs: bone marrow mesenchymal stromal cells. BM-MSC: bone marrow stroma cells. cMET: tyrosine-protein kinase Met. CCL2: C-C motif ligand 2 chemokine. CCL3: C-C motif ligand 3. CSF2: colony Stimulating Factor 2. CXCL12: C-X-C motif chemokine 12. CXCR4: Cys-X-Cys motif chemokine receptor type 4. C57BL/6: B6 mouse strain. CD34: marker of human HSPC. DKK1: Dickkopf-related protein 1. EVs: Extracellular vesicles. GATA2: GATA2 transcription factor. GW4869: neutral sphingomyelinase-2 specific inhibitor HD: healthy donor. Hif-1: Hypoxia-inducible factor 1. HR: hypoxia resistant. HSPC: hematopoietic stem and progenitor cells. IL1b: interleukin 1 beta. IGF1: Insulin-like Growth Factor 1. JNK: c-Jun N-terminal kinases. KITL: KIT ligand. MAPK: mitogen-activated protein kinase. MCP-1: Monocyte chemoattractant protein-1. MDM2: Mouse double minute 2 homolog. MDS: myelodysplastic syndromes. MGUS: monoclonal gammopathy of unknown significance. miR: microRNA. MM: Multiple Myeloma. MPL: myeloproliferative leukemia virus oncogene. MVs: Micro-vesicles. NF-κB: nuclear factor kappa-light-chain-enhancer of activated B cells. NOD-SCID: Nonobese Diabetic-Severe Combined Immunodeficiency. pErk1/2: phosphorylated extracellular-regulated kinase 1/2. peIF4E: phosphorylated eukaryotic translation initiation factor 4E. peIF4GI: phosphorylated eukaryotic translation initiation factor 4GI. pJNK: phosphorylated c-Jun N-terminal kinases. OCN: Osteocalcin. SDF-1: stromal cell-derived factor 1. TI factor: transcription initiation factor. TLR-4: Toll-like receptor 4. ZFP36: zinc finger protein 36. MVs: microvesicles.

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
