# Peer review of "The Role of Bone Marrow Mesenchymal Stem Cell Derived Extracellular Vesicles (MSC-EVs) in Normal and Abnormal Hematopoiesis and Their Therapeutic Potential"

_jcm, 2020, doi:10.3390/jcm9030856_

Round 1

Reviewer 1 Report

The present article entitled "The role of bone marrow mesenchymal stem cell-derived extracellular vesicles (MSC-EVs) in normal and abnormal haematopoiesis and their therapeutic potential." seems to be interesting and gave useful insights about the role of extracellular vesicle in normal and abnormal haematopoiesis involved and their possible clinical use. However, before considering this article to be accepted, it needs to add some aspects that the authors do not consider and review the manuscript to give more precise data

In particular,

Authors have mentioned about several studies. Therefore, it would be much better to put more latest and relevant references to support their statement. For example at the end of the paragraph 3.2. the authors write “Τo the best of our knowledge, there is one study showing that MSCMVs from MDS patients modify CD34+ 178 cell properties, promote cell viability and clonogenic 179 capacity and alter their miRNA and gene expression [62-65]”  indicating 5 references (not one study!) , moreover the reference 62 is about graft versus disease and not about the MDS.

In the table 1 study about human and murine BM-MSC derived EVs’ effects are illustrated. I suggest spitting the table showing studies about human and murine BM-MSCs separately.

The authors do not discuss the EV production for clinical use. I suggest inserting a paragraph about this point and illustrating the different isolation method , the administration way and the regulation to follow for their production for clinical use.  

Therefore, authors are advised to carefully revise the whole manuscript.

Author Response

Responses to the Reviewers' Comments

We would like to thank the Editor and the Reviewers for their constructive criticism which has helped to improve the manuscript. Each comment has been carefully considered point by point.  Responses to the reviewers are as follows.

Reviewer 1

  1. Authors have mentioned about several studies. Therefore, it would be much better to put more latest and relevant references to support their statement. For example at the end of the paragraph 3.2. the authors write “Τo the best of our knowledge, there is one studyshowing that MSCMVs from MDS patients modify CD34+ 178 cell properties, promote cell viability and clonogenic 179 capacity and alter their miRNA and gene expression [62-65]”indicating 5 references (not one study!) , moreover the reference 62 is about graft versus disease and not about the MDS.

Reply. Prompted by this comment by the Reviewer, we have added more references (Ref 76,77,78) (manuscript: Section 3.2, page 5, lines 208-218) about the effect of BM-MSCs in MDS and we corrected the reference 62 in the revised manuscript.

  1. In the table 1 study about human and murine BM-MSC derived EVs’ effects are illustrated. I suggest spitting the table showing studies about human and murine BM-MSCs separately.

Reply. Most of the studies used in this manuscript as also seen in Table 1, include human and murine samples simultaneously, so the splitting  in two different tables as was suggested was difficult. We have  include more references (Ref 69,70 and 73) (manuscript: Section 3.2, page 5, lines 196-197 and 205-206). The same references were also included  in the Table of the revised manuscript.

  1. The authors do not discuss the EV production for clinical use. I suggest inserting a paragraph about this point and illustrating the different isolation method , the administration way and the regulation to follow for their production for clinical use.

Reply. We would like to thank the reviewer for this comment

We have added further information regarding the isolation of EVs and their handling.  (manuscript: Section 2, page 3, lines 103-130). Furthermore, we present to the reviewer a table  with the main isolation methods and their characteristics as seen  below. However, we believe  that the extensive reference to  isolation methods  is out of the scope of this manuscript. Therefore, we haven’t incorporated the table in the main  manuscript but we have included it as a supplemental file.

It has been reported that locally administered EVs may achieve very high local concentrations at target sites. Moreover, intranasal administration of vesicles  has already been tested in mice [1]. However, other types for administration such as intrathecal, intracerebral and intraventricular have not yet been tested [2].

According to the US Food and Drug Administration (FDA) (http://www.fda.gov/downloads/drugs), in the context of clinical trials, the route of administration, the acute and repeated dose, the local toxicity studies with histological evaluation and the route-specific considerations should be taken into consideration [3].

Method

EV yield

Purity

 Scalability

Advantages

Disadvantages

Differential centrifugation (DC)

Used to isolate crude EV/exosome mixtures from conditioned media (CM) by DC at ~100.000 g after the removal of cell debris/intact cells by low-g force centrifugation steps. After an initial low-g force spin MVs can be isolated from CM; purified exosomes can be harvested by a next step centrifugation at 100.000 g [4]

Medium

(high molecular mass protein complexes, sample heterogeneity)

Low

Medium

Commonly used method allowing comparison between studies

Can be combined with concentration methods to produce large quantities

Includes contaminants without additional isolation steps

EVs may aggregate and lose functionality

Pellet can be difficult to resuspend

Density gradient ultracentrifugation (DGC)

Fractionates EVs on the basis of buoyant density using a discontinuous gradient of a sucrose solution. Generally, it results in reduced levels of nonvesicular protein contaminants. DGC does not affect complete separation of MVs and exosomes whose buoyant densities overlap [4, 5].

Medium

(sample loss during fractionation)

High

Low

Commonly used method allowing comparison between studies.

 Some gradient media, for example, iodixanol, impair EV function less than others.

Among the highest purity products

Some media, for example, sucrose, may interfere with EV function

Rotor size limits the total volume that can be processed.

High performance liquid chromatography (size exclusion)

A well-established, high-yield method for purifying functional EVs for tissue regeneration studies [6]. Requires specialized equipment and is time consuming, but the general principles can be applied to simple.

Widely used for isolating EVs from plasma samples and has been adapted for high-throughput clinical samples [5]. This method overcomes many of the problems associated with EV isolation from plasma/serum using DC/DGC [7, 8].

Medium

(loss of small size exosomes)

Medium

High

Ideal for large scale

Shown to preserve therapeutic activity

Magnetic bead isolation

Affinity capture methods for isolating EVs rely on an affinity tag (mAb that targets an EV surface antigen, biospecific peptide [9], or proteoglycan affinity reagents [10]) covalently fused to either magnetic or agarose beads.

In a study AC was reported to be more effective than DG and DGC [4] .

Low

(Ab selection/availability dependent)

High

Low

Highly pure end rapid product

Costly

Depends on knowledge of specific surface markers

Necessity to remove EVs from antibodies or other affinity agents, which may mask molecules required for target selection.

  1. Salama, H.A., et al., Brain delivery of olanzapine by intranasal administration of transfersomal vesicles. J Liposome Res, 2012. 22(4): p. 336-45.
  2. Gyorgy, B., et al., Therapeutic applications of extracellular vesicles: clinical promise and open questions. Annu Rev Pharmacol Toxicol, 2015. 55: p. 439-464.
  3. Fuster-Matanzo, A., et al., Acellular approaches for regenerative medicine: on the verge of clinical trials with extracellular membrane vesicles? Stem Cell Res Ther, 2015. 6: p. 227.
  4. Tauro, B.J., et al., Comparison of ultracentrifugation, density gradient separation, and immunoaffinity capture methods for isolating human colon cancer cell line LIM1863-derived exosomes. Methods, 2012. 56(2): p. 293-304.
  5. Greening, D.W., et al., A protocol for exosome isolation and characterization: evaluation of ultracentrifugation, density-gradient separation, and immunoaffinity capture methods. Methods Mol Biol, 2015. 1295: p. 179-209.
  6. Lai, R.C., et al., Exosome secreted by MSC reduces myocardial ischemia/reperfusion injury. Stem Cell Res, 2010. 4(3): p. 214-22.
  7. Boing, A.N., et al., Single-step isolation of extracellular vesicles by size-exclusion chromatography. J Extracell Vesicles, 2014. 3.
  8. Welton, J.L., et al., Ready-made chromatography columns for extracellular vesicle isolation from plasma. J Extracell Vesicles, 2015. 4: p. 27269.
  9. Ghosh, A., et al., Rapid isolation of extracellular vesicles from cell culture and biological fluids using a synthetic peptide with specific affinity for heat shock proteins. PLoS One, 2014. 9(10): p. e110443.
  10. Christianson, H.C., et al., Cancer cell exosomes depend on cell-surface heparan sulfate proteoglycans for their internalization and functional activity. Proc Natl Acad Sci U S A, 2013. 110(43): p. 17380-5.

Reviewer 2 Report

The review covers a potentially important subject whereby MSC derived EV could become therapeutically relevant in cases where MSC are being considered for therapy.

Overall, scope and detail are appropirate

Among areas of application, GVHD is covered and well annotated. The review might gain in appeal if additional areas could be covered, such as those in line 97. In particular the us of EV for cardiovascular applications would lend itself to expanded coverage. Many articles on the subject area have been published.

Author Response

Responses to the Reviewers' Comments

We would like to thank the Editor and the Reviewers for their constructive criticism which has helped improve the manuscript. Each comment has been carefully considered point by point. Responses to the reviewers are as follows.

Reviewer 2

1.Among areas of application, GVHD is covered and well annotated. The review might gain in appeal if additional areas could be covered, such as those in line 97. In particular the us of EV for cardiovascular applications would lend itself to expanded coverage. Many articles on the subject area have been published.

Reply. We would like to thank the reviewer for this comment.  Except from hematological malignancies, MSC-EVs are involved in cardiovascular, liver, kidney diseases, in wound healing process and in tumor inhibition and might be used as a cell-free therapy. The main roles are described below. However, we think that the role of EVs in the above conditions, is out of the scope of this review which is focused on hematological disorders. Therefore, we think it is better not to incorporate it in the manuscript.

 EVs and Cardiovascular Diseases

Many studies have used MSC-EVs for  Cardiovascular Diseases (CVD) including Myocardial infraction (MI), Ischemia-Reperfusion (IR), injury and coronary artery diseases. Circulating levels of EVs derived from blood vessel cells were found increased in CVD, leading to regard them as prognostic and diagnostic biomarkers. They can also be used as biological vectors, due to their ability to carry and transfer biological information at the injury area [1]. The main role of EVs is to regulate the inflammation, angiogenesis and resolution of the injured tissues via the transfer of inflammatory cytokines and microRNAs from cardiomyocytes to immune cells, endothelial cells and fibroblasts [2]. Previous studies have shown that when MSCs are exposed to hypoxia, they release large amounts of EVs, capable to promote the neoangiogenesis and the recovery of infracted heart via the protection of cells from apoptosis. Another study has demonstrated that the intramyocardial injection of MSC-EVs enhance blood flow recovery and boost the effect of Cardiac Stem Cells (CSCs) in rats. Moreover, the co-culture of CSCs with MSC-EVs increases the proliferation, the migration and the angiogenic potency in vitro, reduces the cardiac fibrosis and improves the cardiac outcome in vivo [3].

5.2 EVs and Liver Diseases

Many studies have demonstrated that EVs are involved in liver diseases such as viral hepatitis, drug-induced injury, alcohol injury, non-alcoholic steatohepatitis and biliary injury [4]. The involvement of EVs in liver diseases is demonstrated by the fact that the composition of miRNAs of EVs from cirrhotic patients is different in contrast to healthy donors. Moreover,       Yin et. al have shown that human umbilical MSCs (huMSCs) inhibit epithelial to mesenchymal transition (EMT), thus ameliorating liver fibrosis. huMSCs-exosomes express glutamine peroxidase 1 (GPX1) which is related to the reduction of hepatic oxidative stress and apoptosis [5].

5.3 EVs and Kidney Diseases

Many studies have demonstrated that MSC-EVs are capable to promote tissue repair and reduce inflammation during Acute Kidney Injuries (AKI). MSC-EVs promote the proliferation of cells and protect them from apoptosis [6]. miRNAs of EVs are responsible for the stimulation of the recipient injured cells for re-entry into the cell cycle [7]. Moreover, the transfer of IGF-1 from EVs to tubular cells may be responsible for renal recovery [8].

5.4 EVs and Wound Healing

During wound healing, EVs are capable to promote the transition of M1 pro-inflammatory to M2 anti-inflammatory macrophages, via the down-regulation of nitric oxide synthase (iNOS), cyclooxygenase (COX)-2, tumor necrosis factor (TNF)-α, interleukin (IL)-1β and monocyte chemoattractant protein (MCP)-1, and the up-regulation of anti-inflammatory cytokines such as IL10. EVs can also suppress the proliferation of T-lymphocytes and activate B-lymphocytes [9]. Moreover, MSC-exosomes are enriched in angiogenic proteins, RNAs and miRNAs, that activate many signaling pathways in endothelial cells. These exosomes also regulate the proliferation and migration of fibroblasts and support the formation of granulation tissue and collagen synthesis [10].

  1. Amosse, J., M.C. Martinez, and S. Le Lay, Extracellular vesicles and cardiovascular disease therapy. Stem Cell Investig, 2017. 4: p. 102.
  2. Chong, S.Y., et al., Extracellular Vesicles in Cardiovascular Diseases: Alternative Biomarker Sources, Therapeutic Agents, and Drug Delivery Carriers. Int J Mol Sci, 2019. 20(13).
  3. Bagno, L., et al., Mesenchymal Stem Cell-Based Therapy for Cardiovascular Disease: Progress and Challenges. Mol Ther, 2018. 26(7): p. 1610-1623.
  4. Fiore, E.J., et al., Taking advantage of the potential of mesenchymal stromal cells in liver regeneration: Cells and extracellular vesicles as therapeutic strategies. World J Gastroenterol, 2018. 24(23): p. 2427-2440.
  5. Yin, S., et al., Human umbilical cord mesenchymal stem cells and exosomes: bioactive ways of tissue injury repair. Am J Transl Res, 2019. 11(3): p. 1230-1240.
  6. Grange, C., et al., Stem Cell-Derived Extracellular Vesicles and Kidney Regeneration. Cells, 2019. 8(10).
  7. Bruno, S., et al., Role of extracellular vesicles in stem cell biology. Am J Physiol Cell Physiol, 2019. 317(2): p. C303-C313.
  8. Tomasoni, S., et al., Transfer of growth factor receptor mRNA via exosomes unravels the regenerative effect of mesenchymal stem cells. Stem Cells Dev, 2013. 22(5): p. 772-80.
  9. Hu, P., et al., Mesenchymal stromal cells-exosomes: a promising cell-free therapeutic tool for wound healing and cutaneous regeneration. Burns Trauma, 2019. 7: p. 38.
  10. Sahoo, S., et al., Exosomes from human CD34(+) stem cells mediate their proangiogenic paracrine activity. Circ Res, 2011. 109(7): p. 724-8.

Round 2

Reviewer 1 Report

The manuscript has been appropriately reviewed for the publication IN JCM.